

# Technical note: Numerical quantification of the mixing states of partially-coated black carbon based on the single-particle soot photometer: Implication for global radiative forcing

Jie Luo[1], Miao Hu[1], Jibing Qiu[2], Kaitao Li[3], Hao He[4], Yuping Sun[4], and Xiulin Geng[1]

[1]College of Communication Engineering, Hangzhou Dianzi University, Hangzhou, Zhejiang, 310018, China.
[2]Institute of Computing Technology, Chinese Academy of Sciences, Beijing 100190, China
[3]School of Information, Space Engineering University, Beijing, 101416, China.
[4]China Jiliang University, College of Energy Environment and Safety Engineering, Hangzhou, Zhejiang, China

**Correspondence:** Jie Luo (luojie@hdu.edu.cn),Miao Hu (miao_hu@hdu.edu.cn)

**Abstract.**

In this work, we have performed a series of numerical investigations on the mixing states of partially-coated black carbon (BC) based on the single-particle soot photometer (SP2). First, we calculated the scattering signal returned from partially-coated BC based on the SP2 measurement, and then the mixing states were determined using Mie theory, where the difference between the determined and "true" mixing states can represent the uncertainties of the SP2 measurement. We found that the SP2 measurement can provide good estimates for small, heavily coated BCs and shows better performance for fully coated BCs. However, the microphysical properties of BCs have a significant impact on the accuracy of the SP2 measurement; sometimes deviations of about -22% to 28% were observed for the determined particle-to-core size ratio ($D_p/D_c$). When considering a size distribution, the error in the effective radius is generally within about -17% to 8.8%. We also investigated the effects of Mie-based models using the SP2 determined and volume-mean $D_p/D_c$ on the radiative effects of partially-coated BC. We found that both Mie models based on the volume mean and SP2 determined mixing states overestimate the absorption enhancement ($E_{abs}$) and direct radiative forcing (DRF) of BC. The Mie model based on the SP2 measurement does not necessarily provide worse estimates of radiative properties, although some errors occur in the determination of the mixing states, since the fraction of the coated core (F) in the particle scale is an important factor affecting $E_{abs}$ and DRF. Sometimes the inaccurate measurements of the mixing states by SP2 would offset the influence of F. Moreover, our results based on the Mie model considering F can significantly improve the estimates for the absorption and DRF of partially-coated BC, although the morphology also has some influence. Therefore, we suggest adding a parameter F to model the radiative effect of BC in climate modeling even when a Mie-based model is used.

## 1 Introduction

Black carbon (BC), the most important absorbing aerosol in the atmosphere, plays an important role in global warming (Bond et al., 2013; Myhre et al., 2013). BC generally absorbs solar radiation strongly from the UV to the near infrared, so it can lead to a positive radiative forcing in the atmosphere (Bond and Bergstrom, 2006; Cross et al., 2010; Petzold and Schönlinner, 2004).





In addition, other chemical compositions can also be deposited on the BC, which leads to an increase in BC absorption due to the "lens effect" (Adachi et al., 2007; Bond et al., 2013; China et al., 2013; Wang et al., 2021a; Peng et al., 2016; Liu et al., 2017b). There are numerous BC aerosols in the atmosphere, and each BC is coated with different amounts of coating materials (Fierce et al., 2020; Wang et al., 2023). Therefore, BC aerosols generally exhibit complex mixing states, which increases the uncertainties in estimating aerosol absorption and radiative forcing. Understanding the mixing states of coated BC is very important for climate modeling.

The single particle soot photometer (SP2), an instrument for measuring the mass of individual BC particles, has recently been widely used to measure mixing states (Schwarz et al., 2006; R. S. Gao and Worsnop, 2007). The SP2 measures scattering of individual particles reflected from a 1064 nm laser, and the mass of the BC core is estimated from the incandescence signal (Moteki and Kondo, 2008; Wu et al., 2023). Based on an assumed BC mass density, we can calculate the mass-equivalent diameter of the BC cores. In the SP2 algorithm, the total diameter of the particle is usually determined using the complex leading-edge only (LEO) technique, which generally estimates the particle diameter based on the scattering signal (R. S. Gao and Worsnop, 2007; Naseri et al., 2023). However, the SP2 calculation assumes that the particles are spherical and the BC cores are fully coated.

BC aerosols usually have a complex morphology and are often partially-coated (Adachi et al., 2007; China et al., 2013; Wang et al., 2017). Although most researchers have recognized that simplifying the microphysical properties of BC aerosols can lead to inaccurate determination of mixing states, there is still a lack of quantification of the effects of microphysical properties. In some studies, fully enveloped models have been used to determine the effects of BC morphology on the determination of mixing states (Wu et al., 2023; Liu et al., 2023). However, BC is often partially-coated, and the absorption of partially-coated BC is more complex than that of fully coated BC. The absorption of partially-coated BC is determined not only by the ratio of the core volume to the total particle, but also by the ratio of the volume of the coated BC cores to the total volume of the BC cores. In this work, we attempt to quantify the effects of the microphysical properties on the determination of the mixing states of partially-coated BC and present the implications for global radiative forcing calculations. In this work, we mainly focus on answering the following two questions:

– How the microphysical properties of partially-coated BC affect the SP2 retrieval of mixing states?

– How the inaccurate retrieval of mixing states affect the calculations of aerosol absorption and radiative forcing?

## 2 Methods

### 2.1 BC morphological model

Freshly emitted BC is commonly composed of numerous near-spherical monomers, and exhibits fluffy structures (China et al., 2013; Chakrabarty et al., 2006; Wentzel et al., 2003b). Previous studies have shown that the fractal law can greatly characterize



the shape of fresh BC (Sorensen, 2001; Heinson et al., 2017; Luo et al., 2021a, b):

$$N_s = k_0 (\frac{R_g}{R})^{D_f} \tag{1}$$

where $N_s$, $k_0$ and $D_f$ represent monomer number, fractal prefactor and fractal dimension, respectively; $R$ and $R_g$ are the radius
of the monomer and the gyration radius, respectively.

$D_f$ and $k_0$ are two important parameters that reflect the compactness and anisotropy of the particles (Sorensen, 2001; Heinson et al., 2017). For a solid $k_0$, the aggregate generally has a more compact structure if it has a larger $D_f$ (Skorupski et al., 2014; Thouy and Jullien, 1994). It is well known that fresh BC has a fluffy structure, and both the numerical simulations and experiments show an approximate $D_f$ of 1.52 – 1.94 in different combustion sources (Sorensen, 2001; Dhaubhadel et al., 2006; China et al., 2013; Chakrabarty et al., 2006; Wentzel et al., 2003b; China et al., 2014). In this work, we used a typical $D_f$ of 1.8 to represent the fresh BC according to Sorensen (2001). As the particle ages, BC may collapse to a more compact structure (Zhang et al., 2008; Lack et al., 2014; Peng et al., 2016; Bhandari et al., 2019). A larger $D_f$ has often been used to represent BC with more compact structure (Liu and Mishchenko, 2005; Kanngießer and Kahnert, 2018; Cheng et al., 2014; Luo et al., 2019, 2023, 2024). Previous studies have shown that aged BC can sometimes have a $D_f$ of about 2.3 – 2.6, assuming a fractal structure for aged BC (Adachi et al., 2010, 2007; Chen et al., 2016). In this work, we have assumed a $D_f$ of 2.6 for compact BC. $k_0$ is also varied in different regions, but its influence on the optical properties is relatively small, and we only consider a typical value of 1.2 in this work. The BC monomer radius is generally varied in a range of about 8 – 57 nm (Lee et al., 2002; Wentzel et al., 2003a; Mikhailov et al., 2006; Adachi et al., 2007, 2010). However, Kahnert and Kanngießer (2020) has further shown that the typical monomer radius is generally in the range of 10 to 25 nm, and we used an average radius of 20 nm. The monomer number was assumed to be in the range of 5 to 1000, which covers most of the observed BC aerosols.

In the atmosphere, BC can also be mixed internally with other chemical compositions (Adachi et al., 2007; Wang et al., 2021c, b). A special type of internally mixed BC is the partially-coated BC, where only part of the monomers are coated. Not only the BC volume fraction but also the fraction of coated monomers would significantly affect the aerosol absorption. In this work, we mainly consider this type of BC aerosols.

We considered both the effects of BC volume fraction ($f_{BC}$) of the fraction of coated monomers (F). $f_{BC}$ is represented by the ratio of the volume of BC cores to the total particle:

$$f_{BC} = \frac{V_{BC}}{V_{particle}} \tag{2}$$

where $V_{BC}$ and $V_{particle}$ are the volume of BC core and total particle, respectively.

The particle to core diameter ratio ($D_p/D_c$) is used to represent the ratio of particle diameter to core diameter:

$$D_p/D_c = \frac{d_{particle}}{d_{BC}} = (\frac{1}{f_{BC}})^{1/3} \tag{3}$$

where $d_{particle}$ and $d_{BC}$ are the volume-mean diameter of the total particle and BC core, respectively.

The fraction of coated monomers was represented by the ratio of the volume of coated BC core monomers to total BC core monomers:





$$F = \frac{V_{BC\ inside}}{V_{BC}} \tag{4}$$

where $V_{BC\ inside}$ represents the volume of BC core monomers inside the coating shell, and $V_{BC}$ represent the volume of the total BC core monomers.

We used the tunable algorithm developed by Woźniak (2012), which accurately reproduces the fractal law, to generate the fractal BC aggregates. Then, the coating materials were added similarly to previous studies (Zhang et al., 2018; Luo et al., 2023, 2024). We assumed that the coating structure is spherical. If we know the BC volume fraction ($f_{BC}$) and F, the spherical shell can be determined. Then, like Zhang et al. (2018), we moved the BC core to find a position where F is satisfied. We moved the BC monomers intersecting with the coating shell out of the coating to use the efficient numerical method, the multiple-sphere T-matrix (MSTM) method (Mackowski and Mishchenko, 2011; Mackowski, 2022), which is only suitable for calculating the optical properties of spheres without overlap. However, previous studies have shown that the motion has no significant effect on the optical properties (Liu et al., 2017a). The typical morphologies are shown in Figure 1. We must clarify that the real coating structure may be more complex than assumed in this work. However, we mainly focus on the effects of the proportion of coated BC monomers and use a typical morphology as an example to illustrate how the mixing states of SP2 are affected by the microphysical properties of BC. The comparison of BC with spherical coating and more complex coating is shown in other studies (Luo et al., 2019).

In this work, we reflect BC with different aging status by the following configurations: (1) fluffy BC aggregates associated with the coating materials, which can be used to represent the fresh BC since they are not internally mixed; (2) compact BC aggregates associated with the coating materials, which is a type of aged BC; (3) Fluffy BC aggregates partially-coated with coating materials, where F and $D_p/D_c$ increase with atmospheric aging; (3) Compact BC aggregates partially-coated with coating materials, where F and $D_p/D_c$ increase with atmospheric aging; (4) Compact BC aggregates fully embedded in the coating shell, representing highly aged BC. SP2 assumed a core-shell BC structure representing only the highly aged BC in determining the indicated mixing. We are trying to better understand the effects microphysical properties at different aging states using the above configurations.

## 2.2 Calculating of the optical properties of partially-coated BC

The MSTM method, which has an advantage in calculating the optical properties of spheres without overlaps, was used to calculate the optical properties of individual BC aerosols. The MSTM method requires the information of the refractive index and the position of the spheres as input. Bond and Bergstrom (2006) suggested that the refractive index of BC does not vary significantly from ultraviolet (UV) to near infrared (NIR) wavelengths and proposed to use a refractive index of about $1.95 + 0.79i$ at 550 nm. In this work, a fixed BC refractive index $1.95 + 0.79i$ was used for all wavelengths considered in this work. A non-absorbing organic carbon (OC) shell was assumed for the coating materials and the OC refractive index was set to $1.55 + 0i$ (Schnaiter et al., 2005; Bond and Bergstrom, 2006).





## 2.3 Retrieval of mixing states of partially-coated BC

SP2 determines the particle size based on the scattering signal, which is generally proportional to the scattering cross-sections. Previous studies have shown that SP2 generally detects the scattering signal at scattering angles ($\Theta$) of $45° \pm 32°$ and $135° \pm$

115    $32°$ (Moteki and Kondo, 2008; Wu et al., 2023). We used the following equation to represent the scattering signal returned by the coated BC:

$$I_{sca} \propto C_{sca\_det} = C_{sca} \sum_{\Theta=0°}^{180°} \omega(\Theta)\sin\Theta P_{11}(\Theta) \tag{5}$$

where $I_{sca}$ represents the detected scattering signal, which is proportional to the detected scattering cross-section at the assumed angles ($C_{sca\_det}$); $C_{sca}$ and $P_{11}$ represent the scattering cross-section and the phase function of the coated BC, respectively. $\omega(\Theta)$ is the weighting function to represent the scattering distribution at different scattering angles, and we used the same

120    $\omega(\Theta)$ as Wu et al. (2023). After calculating $C_{sca\_det}$ of partially-coated BC, the diameter of coated BC is determined based on Mie theory by finding the diameter with the best agreement with $C_{sca\_det}$. In this work, the Mie calculations were performed by PyMieScatt (Sumlin et al., 2018).

Retrieved $D_p/D_c$ ($D_p/D_c\_retr$) is determined by:

$$D_p/D_c\_retr = \frac{d_{particle\_retr}}{d_{BC}} \tag{6}$$

where $d_{particle\_retr}$ represents the particle diameter determined on the basis of Mie theory.

## 125   2.4 Retrieved effective radius of BC with size distributions

There are numerous BC aerosols in the atmosphere, and we have also investigated the determined effective radius of BC with size distributions. We assume a lognormal size distribution for the BC core:

$$n(r_v) = \frac{1}{\sqrt{2\pi}r_v\ln(\sigma_g)}\exp\left[-\left(\frac{\ln(r_v)-\ln(r_g)}{\sqrt{2}\ln(\sigma_g)}\right)^2\right] \tag{7}$$

where $r_g$ and $\sigma_g$ are the geometric mean radius and geometric standard deviation, respectively.

130    The effective radius ($R_{eff}$) of coated BC is calculated with:

$$R_{eff} = \frac{\int_{r_{min}}^{r_{max}} r_v \pi r_v^2 n(r_v)dr_v}{\int_{r_{min}}^{r_{max}} \pi r_v^2 n(r_v)dr_v} \tag{8}$$

## 2.5 Calculating the absorption enhancement of BC with different mixing states

To evaluate the effects of retrieval uncertainties on radiative forcing, we calculated the bulk absorption enhancement of partially-coated BC and the enhancement calculated with Mie theory based on the retrieved mixing states. Considering the



size distribution of the BC core, the absorption enhancement of partially-coated BC ($E_{abs\_partially\_coated}$)is calculated as follows:

$$E_{abs\_partially\_coated} = \frac{\int_{r_{min}}^{r_{max}} C_{abs\_partially\_coated}(r_v)n(r_v)dr_v}{\int_{r_{min}}^{r_{max}} C_{abs\_core}(r_v)n(r_v)dr_v} \tag{9}$$

where $C_{abs\_partially\_coated}$ and $C_{abs\_core}$ represent the absorption cross-section of partially-coated BC and BC core, respectively. The absorption enhancement calculated using Mie theory with the volume-mean diameter can be express as:

$$E_{abs\_core\_shell} = \frac{\int_{r_{min}}^{r_{max}} C_{abs\_coated\_Mie}(r_v)n(r_v)dr_v}{\int_{r_{min}}^{r_{max}} C_{abs\_Mie\_core}(r_v)n(r_v)dr_v} \tag{10}$$

where $C_{abs\_coated\_Mie}$ and $C_{abs\_Mie\_core}$ represent the Mie-based absorption cross-section of coated BC and BC core, respectively.

The absorption enhancement calculated using Mie theory with the retrieved mixing states ($E_{abs\_core\_shell\_retrieved}$) can be express as:

$$E_{abs\_core\_shell\_retrieved} = \frac{\int_{r_{min}}^{r_{max}} C_{abs\_coated\_Mie\_retrieved}(r_v)n(r_v)dr_v}{\int_{r_{min}}^{r_{max}} C_{abs\_Mie\_core}(r_v)n(r_v)dr_v} \tag{11}$$

where $C_{abs\_coated\_Mie\_retrieved}$ represents the Mie-based absorption cross-section of coated BC based on the retrieved $D_p/D_c$.

## 2.6 Estimating the effects of SP2 retrieval on global radiative effect

In this work, we have investigated the effects of Mie-based models using the mixing states determined in SP2 on global absorption and radiative forcing. First, a global chemistry transport model, the Goddard Earth Observing System with chemistry (GEOS-Chem) model (Bey et al., 2001; Eastham et al., 2018), was used to simulate the BC distribution. The GEOS-Chem simulation was identical to Luo et al. (2024), and a full-chemistry standard simulation was performed for the year 2016. The spatial resolution was set to $4° \times 5°$ and GEOS-Chem was built with 47 vertical layers. Anthropogenic emissions were calculated based on the Community Emissions Data System (CEDS), and emissions from biomass burning were based on the Global Fire Emissions Database (GFED4) inventory (Randerson et al., 2018). The Model of Emissions of Gases and Aerosols from Nature Version 2.1 (MEGAN 2.1) was used to generate the biogenic emissions(Guenther et al., 2012). GEOS-Chem was performed on the basis of MERRA-2 (second Modern-Era Retrospective analysis for Research and Applications) with assimilated meteorology(Molod et al., 2015; Gelaro et al., 2017). After the GEOS-Chem simulation is completed, we can determine the global distribution of BC concentrations. We have taken the temporal average for the BC concentrations over the entire year 2016. Then we can calculate the aerosol optical absorption depth based on the BC mass concentration:

$$AAOD_{BC} = MAC_{BC} \times E_{abs} \times C_{BC\_column} \tag{12}$$

where $C_{BC\_column}$ is the column mass concentration of BC; $MAC_{BC}$ represents the mass cross-section of the BC core; $E_{abs}$ is the absorption enhancement of coated BC. In principle, $MAC_{BC}$ can be calculated using MSTM or Mie theory. However, an





inconsistency between the modeled and measured $\mathrm{MAC_{BC}}$ has been found, and most models underestimate the MAC of BC based on the measured mass density and refractive index (Kahnert, 2010; Luo et al., 2018; Fengshan Liu and Corbin, 2020). In

this work, we used a $\mathrm{MAC_{BC}}$ of $7.5 \pm 1.2 \ \mathrm{m^2 g^{-1}}$ proposed by Bond and Bergstrom (2006).

     The direct radiative forcing (DRF) is estimated based on a simple method similar to Kelesidis et al. (2022). Based on the values proposed in Bond et al. (2013), Kelesidis et al. (2022) suggested to use an average absorption forcing efficiency of 170 $\pm$ 43 $\mathrm{Wm^{-2}}$ /AAOD. We estimated the DRF by multiplying the estimated AAOD by 170 $\pm$ 43 $\mathrm{Wm^{-2}}$ /AAOD.

## 3    Results

### 3.1    The effects of BC microphysical properties on the SP2 retrieved mixing states

Figure 2 shows the retrieved mixing states of BC with different microphysical properties. It can be seen that the determined $\mathrm{D_p/D_c}$ does not deviate significantly from the "true" $\mathrm{D_p/D_c}$ if the diameter of the BC core size ($\mathrm{D_c}$) is small and the "true" $\mathrm{D_p/D_c}$ is large (e.g. $\mathrm{D_p/D_c}$ = 4.6, 2.71). The deviations are generally within 1%. The possible reason is that the large $\mathrm{D_p/D_c}$ and a small BC core cause an overall structure close to the structure of coating materials. The particle structure of small,

heavily coated BC is nearly spherical, so the SP2 determination can provide a reasonable estimate. It should be noted that a spherical coating structure was assumed in this work, while actual coated BC may have a more complex structure, which may be a drawback of this study. However, we mainly used a typical morphology as an example to illustrate how the mixing states of SP2 are affected by the microphysical properties of BC, and more complex morphologies may be considered in the future. As the size of the BC core increases, the deviation between "real" and determined $\mathrm{D_p/D_c}$ increases, and the relative error can

sometimes reach about 28%. In particular, the determined $\mathrm{D_p/D_c}$ are randomly distributed if both $\mathrm{D_c}$ and $\mathrm{D_p/D_c}$ are large. This is because the total particle size is extremely larger when both $\mathrm{D_c}$ and $\mathrm{D_p/D_c}$ are larger, and at this point the scattering cross-section becomes insensitive to the particle size.

     For smaller "true" values $\mathrm{D_p/D_c}$, some obvious deviations are detected, even if $\mathrm{D_c}$ is small. The relative error between the "true" and the determined $\mathrm{D_p/D_c}$ is generally in a range of about -22% – 8%. BC with larger $\mathrm{D_f}$ generally show larger

determined $\mathrm{D_p/D_c}$. This is due to the fact that more compact fractal BC aggregates would lead to a larger scattering cross-section and thus to a larger retrieved $\mathrm{D_p/D_c}$. For the fully embedded BC (F = 1), the configuration of "Outer" and "Centre" has a significant impact on the retrieved $\mathrm{D_p/D_c}$ when $\mathrm{D_p/D_c}$ is large (e.g. $\mathrm{D_p/D_c}$ = 4.64). In general, SP2 retrieval can provide a reasonable estimate for the $\mathrm{D_p/D_c}$ ofthe "centre" configuration if $\mathrm{D_c}$ is not large. The relative error between SP2 retrieval and "real" $\mathrm{D_p/D_c}$ is generally within 3% for our "centre" configurations when $\mathrm{D_c}$ is less than 300 nm. However, the relative

error between SP2 retrieval and "real" $\mathrm{D_p/D_c}$ can sometimes reach up to 8% for "outer" configurations, which means that the positions of the BC cores can affect the SP2 retrieval accuracy.

     Figure 3 – 4 shows the discrepancy between the determined and the "Ture" effective radius of coated BC with typical size distributions. We found that SP2 overestimates the "true" effective radius for both flaky and compact BC when BC is heavily coated (e.g. $\mathrm{D_p/D_c}$ = 4.64). The reason for this phenomenon could be that the core-shell-sphere model underestimates

the scattering cross-section of heavily coated BC, so a larger size is required to fit the scattering cross-section of partially-




coated BC. Moreover, the overestimation becomes more obvious as $r_g$ and $\sigma_g$ increase. In general, SP2 retrievals lead to an overestimation of about 0 to 8.8% for heavily coated BC. However, for BC with thin coatings (e.g. $D_p/D_c = 1.49$), the opposite phenomenon was found and the SP2 generally underestimates the "true" effective radius. The scattering cross-section of thin coated BC is generally overestimated by a spherical assumption, resulting in a smaller retrieved size. The phenomenon is similar

for BC with a fluffy and compact core. In contrast, SP2 determination significantly underestimates the size of thin-coated BC with a fluffy core. Sometimes the SP2 determination can lead to an underestimation of about 17% for fluffy thin-cored BC (e.g. $D_f = 1.8$, $D_p/D_c = 1.49$), while the underestimation for compact thin-cored BC is within 11% (e.g. $D_f = 2.6$, $D_p/D_c = 1.49$). The reason for this is that the core-shell sphere overestimates the scattering cross-section of thin-coated BC aggregates, leading to an underestimation of the effective BC radius, and that thin-coated, fluffier BC generally has a smaller scattering

cross-section, so that the SP2 determination for fluffier BC provides a more significant underestimation.

Figure 5 shows the comparison of $E_{abs}$ of BC with different models, where "partially-coated" represents the $E_{abs}$ partially-coated BC with volume-averaged size; "Mie $E_{abs}$ based on SP2 retrieval" represents the $E_{abs}$ of BC calculated with Mie theory based on SP2 retrieval size; "Mie $E_{abs}$ based on volume-averaged size" represents the $E_{abs}$ of BC calculated with Mie theory based on volume-averaged size. As shown in Figure 5, the core-shell Mie theory significantly overestimates the $E_{abs}$

of partially-coated BC, since the core-shell Mie theory assumes a fully coated structure. In general, Mie-based $E_{abs}$ with SP2 mixed states lead to a relatively lower overestimation than with the "true" mixed states. However, we don't expect a large difference between the Mie-based $E_{abs}$ with SP2 retrieval and the "true" mixing states for fluffy BC (i.e. $D_f = 1.8$), and the difference is generally within 7%. However, sometimes the difference between the Mie-based $E_{abs}$ with SP2 retrieval and the "true" mixing states for compact BC (e.g. $F = 0$, $D_f = 2.6$, $D_p/D_c = 1.49$) can reach about 22%.

## 3.2 The impact of SP2 retrieval on global radiative effects

Figure 6 compares the global mean AAOD calculated with different configurations, where the meaning of the different configurations is given in Table 1, and "M" stands for Mie theory with the retrieved SP2 size; "M-T" stands for Mie theory with the volume mean size. Previous studies have shown that the overall mean value of AAOD is generally about 0.001 to 0.0029 when a pure BC model is used (Kelesidis et al., 2022; Sand et al., 2021; Kim et al., 2008; Chung and Seinfeld, 2002; Schulz

et al., 2006; Textor et al., 2006). However, considering the enhancement of coating materials, the global mean AAOD value could be about 0.0014 to 0.0046. Our simulated global mean AAOD value is generally in the range of 0.0011 to 0.0043, which is generally consistent with values reported in previous studies. We found that the global mean AAOD calculated using Mie theory sometimes reaches about two times. This means that a fully coated Mie theory can lead to a large overestimation of AAOD. We found that the "M" and "M-T" configurations have similar accuracy when the $D_p/D_c$ is large due to an overall

spherical morphology. However, significant differences are observed when $D_p/D_c$ is large. The overestimation of AAOD is generally mitigated by using the size obtained with SP2 instead of the mean volume size when F is small (e.g. $F = 0.05$), while sometimes the opposite is observed when F is relatively large (e.g. $F = 0.2$).

Figures 7 – 8 show the estimated DRF comparison with different configurations. citeRN39 showed that coating materials have on average 6.9 times the BC core volume, which corresponds to a typical $D_p/D_c$ of 1.992. Therefore, we have chosen two





**Table 1.** Case configurations of this work, as shown in Figure 6 and Figure 11.

| A | B | C | D | E | F | H | I |
|---|---|---|---|---|---|---|---|
| F = 0.05 | F = 0.05 | F = 0.05 | F = 0.05 | F = 0.2 | F = 0.2 | F = 0.2 | F = 0.2 |
| $f_{BC}$ =10% | $f_{BC}$ =30% | $f_{BC}$ =10% | $f_{BC}$ =30% | $f_{BC}$ =10% | $f_{BC}$ =20% | $f_{BC}$ =10% | $f_{BC}$ =20% |
| $D_f$ = 1.8 | $D_f$ = 1.8 | $D_f$ = 2.6 | $D_f$ = 2.6 | $D_f$ = 1.8 | $D_f$ = 1.8 | $D_f$ = 2.6 | $D_f$ = 2.6 |

typical $D_p/D_c$ of 2.15 and 1.71 for illustration, which can generally reflect the real atmosphere. As can be seen in Figure 8, the DRF of BC is significantly affected by the morphology and F. As expected, for fixed $D_p/D_c$, the DRF generally increases with increasing F. In addition, Figures 7 – 8 clearly show that both the core-shell model with "True" and the SP2 model significantly overestimate the DRF of partially-coated BC. The DRF difference between the Mie model and the model for partially-coated BC is even comparable to the DRF of partially-coated BC itself. Especially in some regions (e.g. East Asia),

the core-shell assumption can overestimate the DRF by more than 10 W/m$^2$ for partially-coated BC with an F of 0.05 and 0.2. When the BC core is fully coated, the overestimation is relatively attenuated (e.g. F = 1, $D_f$ = 2.6, Figure 8), and the overestimation is generally within 1 W/m$^2$, even in Esat Asia. This is easy to understand since the core-shell model can only represent the fully coated BC. Our simulations show that the DRF difference between the Mie size determined with SP2 and the mean volume size is significantly affected by the $D_p/D_c$. As shown in Figure 7, the DRF difference between the "CS True"

and "SP2" configurations can reach over 2 W/m$^2$ in some regions (e.g. East Asia) when $D_p/D_c$ = 1.71, which is due to the inaccurate determination of SP2. However, the DRF difference between the "CS True" and "SP2" configurations is ignorable when $D_p/D_c$ reaches 2.15, and the difference is generally within 0.1 W/m$^2$ (see Figure 7). This is because the core-shell structure can generally provide a reasonable estimate for the scattering cross-sections of heavily coated BC, whose overall structure is nearly spherical.

**3.3 The importance of F in estimating radiative effects**

In the calculations above, we have found that the core-shell Mie theory can provide rather inaccurate estimates for the partially-coated BC, using both the retrieved SP2 and the "real" mixing states. However, we found that the effects of core structure are relatively small compared to the proportion of coated core when comparing the cases $D_f$ = 1.8, $D_f$ = 2.6 and core-shell structure. In application, Mie theory is still a reasonable choice to model aerosol absorption as it is difficult to determine which specific

model should be used since BC aerosols have different morphologies. In application, the goal may be to investigate how the Mie model could provide better estimates for the absorption enhancement of partially-coated BC. As shown in Figure 9, we use an uncoated Mie model and a coated Mie model to represent the absorption enhancement of partially-coated BC. The absorption cross-section of the Mie model for individual particles considering the effects of F is expressed as follows:

$$C_{abs\_coated}(r_v, r_p) = C_{abs\_coated}(r_v \times F^{1/3}, r_p) + C_{abs\_core}(r_v \times (1-F)^{1/3}) \qquad (13)$$





$$C_{abs\_core}(r_v) = C_{abs\_core}(r_v \times F^{1/3})) + C_{abs\_core}(r_v \times (1 - F)^{1/3}) \quad (14)$$

where $C_{abs\_coated}(r_v, r_p)$ represents the absorption cross-section of coated BC with a core radius of $r_v$ and a shell radius of $r_p$; $C_{abs\_core}(r)$ represents the absorption cross-section of coated BC with a core radius of $r$. It should be noted that the core morphology can also influence the absorption gain. However, we assume that $D_p/D_c$ and F are the two main factors affecting the absorption gain, so our assumption is reasonable for practical application and the effects of core structure will be investigated in other studies.

It is clear from Figure 10 that the performance of the model in estimating the absorption gain of partially-coated BC is significantly improved by adding an F to characterize the absorption at the particle level. Even though the difference between the Mie model and the morphological model is sometimes still about 20%, the performance is significantly better compared to the traditional Mie model. This suggests that our simplified Mie model could be a substitute for the traditional model if we know the F at the particle scale. Especially in field measurements, a fully coated model often overestimates the absorption gain of BC. Many studies have attributed this to the non-uniform mixing states at the particle level. However, more recent studies have also shown that the absorption gain can be overestimated even when the effects of non-uniform mixing conditions are taken into account. The reasons for this probably lie in the partially-coated BC. However, if we use a morphologically realistic model, the calculations are quite computationally intensive, and it is difficult to perform queries to find the F based on the optical measurements and incorporate them into large models (e.g., climate models). Therefore, a simplified model such as our model is a good choice, even if some reasonable error can be introduced.

The global simulations (see Figure 11 – 12) also show that our simplified model can improve the performance of the traditional Mie models when F is taken into account. The global mean AAOD difference between the improved Mie model and the morphologically realistic model is generally within 20%. Moreover, we see that the global AAOD difference calculated using "True" and SP2 computed mixture states is weakened by considering the effects of F and the difference becomes ignorable. A comparison of Figure 12 and Figure 7 also shows that the overestimation of the DRF with the Mie model is significantly weakened by considering the effects of F and the overestimation of the Mie model is generally within +2 W/m$^2$, even in highly polluted regions (e.g. East Asis), which is slightly lower than in Figure 7. The DRF difference between the Mie model with "True" and the mixing states determined with SP2 has also decreased to below 0.1 W/m$^2$. Therefore, considering F at the particle scale is very important for improving the model performance in estimating the radiation effects, even when using a simplified model. We therefore suggest considering the effects of F in climate modeling, as this is an important factor affecting accuracy.

### 3.4 Implication for modeling the global direct radiative effect

The mixing states of BC aerosols are one of the most important factors for the uncertainties of climate modeling. In particular, the mixing states would significantly influence the direct radiative effect. In the measurements, the layer thickness parameters (e.g. $D_p/D_c$) are often used to represent the mixing state. In this work, we conducted a series of numerical investigations on




the effects of the microphysical properties of BC on the mixing states determined by SP2 measurements. We found that the accuracy of SP2 is significantly affected by the microphysical properties of BC, and sometimes a deviation of about -22% to 28% was observed when BC has a partially-coated morphology. This means that the radiation calculations based on SP2

determination would lead to additional uncertainties. However, our results also show that both the core-shell models based on the SP2 measurement and the volume-average model lead to inaccurate estimates of the radiative properties, while the Mie model based on the SP2 measurement does not necessarily provide worse estimates of the radiative properties. The reason for this is that F is the main factor except for $D_p/D_c$ affecting the radiation effects are the F, while the core-shell model assumes a fully-coated structure. Sometimes the inaccurate measurements of the mixing states by SP2 would offset the influence of F.

Moreover, our results based on the Mie model considering F can significantly improve the estimates for the absorption and DRF of partially caotized BC, even though the morphology also has some influence. Therefore, we suggest adding a parameter F to model the radiative effect of BC in climate modeling even if a Mie-based model is used.

## 4 Summary and conclusions

The mixing states of BC play an important role in climate impacts due to the "lensing effect". The SP2 is a commonly used

instrument to measure BC concentrations, and it is also commonly used to measure mixture states. However, the mixing states measured with the SP2 are mainly based on the optical retrieval method according to Mie theory. The aerosol is assumed to be spherical, while BC aerosols sometimes have a complex morphology. In this work, a special type of BC aerosol, partially-coated BC, was considered because its absorption is influenced not only by the mixing states but also by F. We first calculate the scattering signal returned from partially-coated BC based on the SP2 measurement, and then the mixing states were retrieved

using Mie theory, and the difference between the retrieved and "true" mixing states can be the uncertainties ofthe SP2 - Represent measurement. Considering different configurations, we investigated the effects of the microphysical properties of BC on the SP2 measurement. We found that the SP2 measurement can provide good estimates for small, heavily coated BCs and shows better performance for fully coated BCs. However, the scattering cross-section is not sensitive to the particle size when the particle size reaches an upper limit, so the SP2 retrieval may produce inaccurate estimates for large heavily coated

BC due to the overall extreme large particle size, sometimes the errors can reach about 28%. Furthermore, for thinly coated BC, the SP2 calculation sometimes results in a relative error of about -22% due to the effects of non-spherical morphology. However, when considering a size distribution, the error in effective radius is generally within about -17% to 8.8%.

We also evaluate the performance of the Mie model with "real" and SP2 mixed states. We found that the Mie-based model can lead to inaccurate estimates for the absorption enhancement and DRF of partially-coated BC, and the Mie-based models

can generally only represent the fully coated BC. Sometimes the Mie-based global mean AAOD can reach 2 times the value of partially-coated BC, and in some regions (e.g. East Asia) the DRF difference can reach +10 W/m$^2$. The difference between the Mie-based AAOD and the DRF is significantly affected by $D_p/D_c$, and the difference is generally ignorable when $D_p/D_c$ is large for a typical size distribution. However, the DRF difference can reach over 2 W/m$^2$ when $D_p/D_c$ is relatively small. The Mie-based model generally provides a better estimate for fully coated BC than for partially-coated BC. We show that the



performance of the model is significantly improved to reproduce the absorption enhancement and radiative effects of partially-coated BC by including another parameter, F, in the Mie calculations. Therefore, we propose to consider not only the effects of mixing states ($D_p/D_c$) but also the effects of F for BC aerosols in climate modeling, as recent studies have shown that the Mie-based $E_{abs}$ sometimes still overestimates the measured values, even if the non-uniform mixing conditions are taken into account.

*Acknowledgements.* This work was financially supported by the National Natural Science Foundation of China (Grant No. 42305148, 42175146) and the primary Research and Development Plan of Zhejiang Province (Grant No. 2023C03014, 2024C03252).



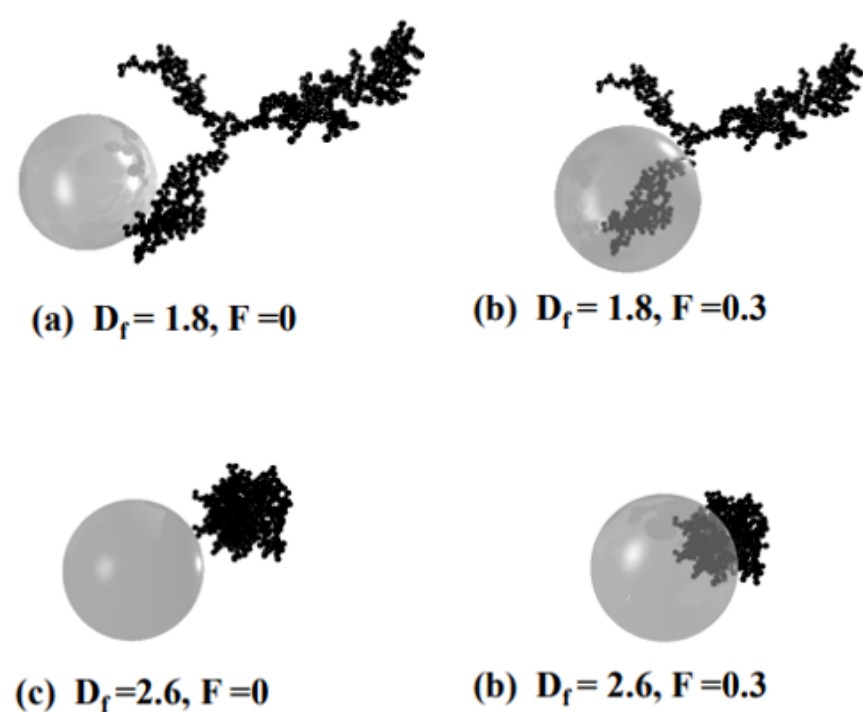

**Figure 1.** Typical morphologies of coated BC considered in this work.



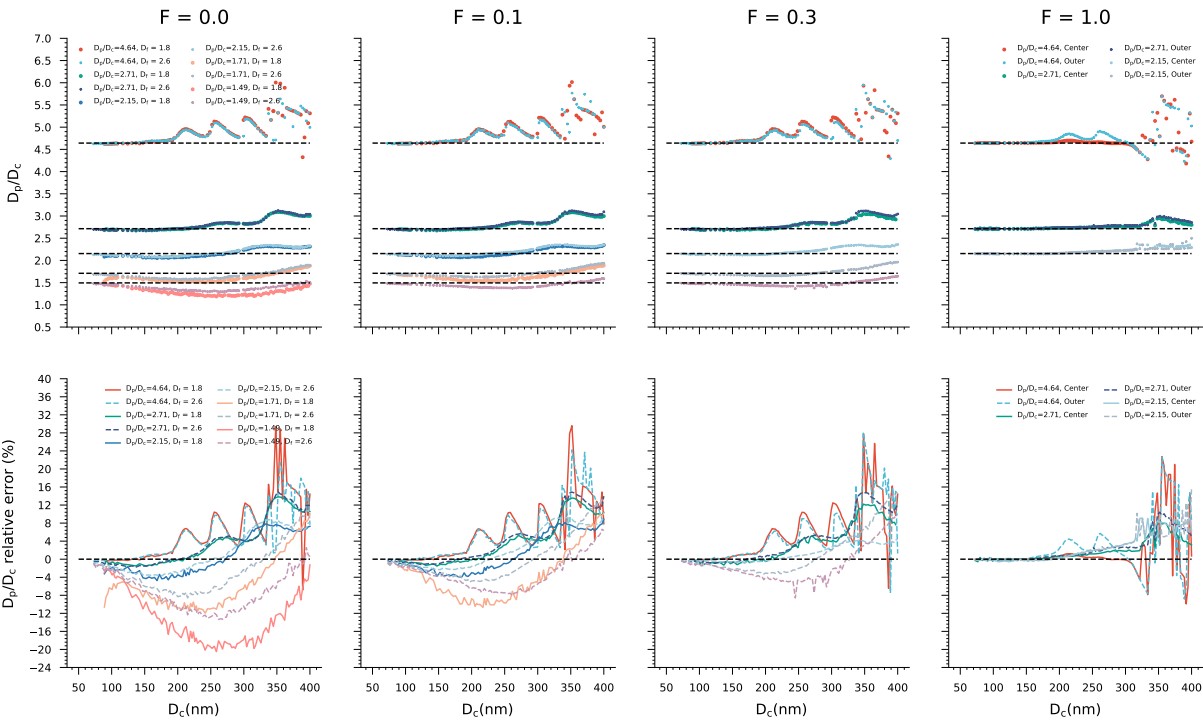

**Figure 2.** The retrieved $D_p/D_c$ of BC with different microphysical properties. SP2 retrieval can generally provide relatively good estimates when the BC kernel size ($D_c$) is small and the "true" $D_p/D_c$ is large. In other cases, however, the accuracy of the SP2 determination is significantly influenced by the microphysical properties of the BC.





**Figure 3.** The difference between SP2 retrieved and "True" effective radius ($R_{eff}$), where $D_f = 1.8$.



**Figure 4.** Similar to Figure 3, but for $D_f = 2.6$.



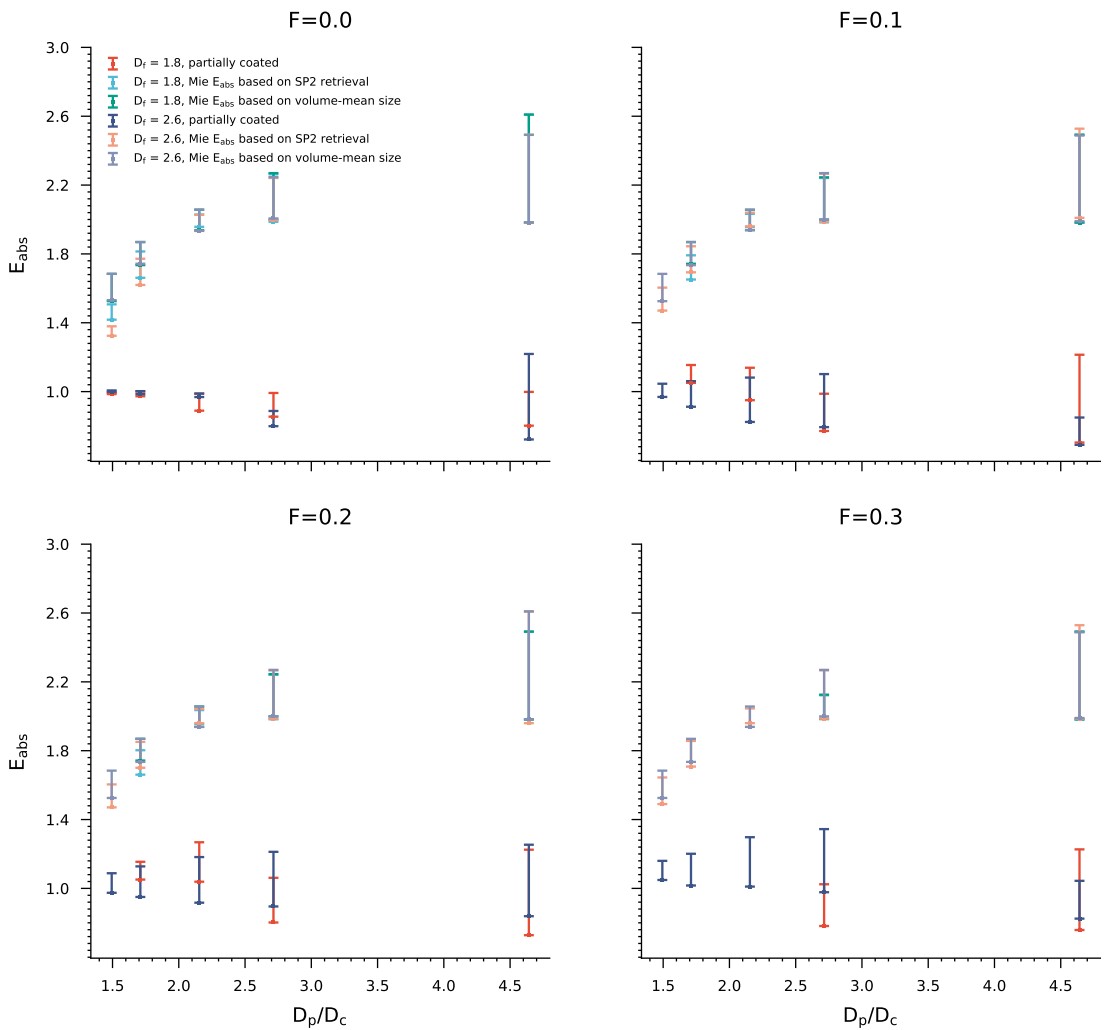

**Figure 5.** The comparison of $E_{abs}$ modeled with morphologically realistic models, Mie models based on the SP2 retrieved and "true" mixing states. Mie-based models significantly overestimate the $E_{abs}$ of partially-coated BC due to the assumption of fully coated BC.



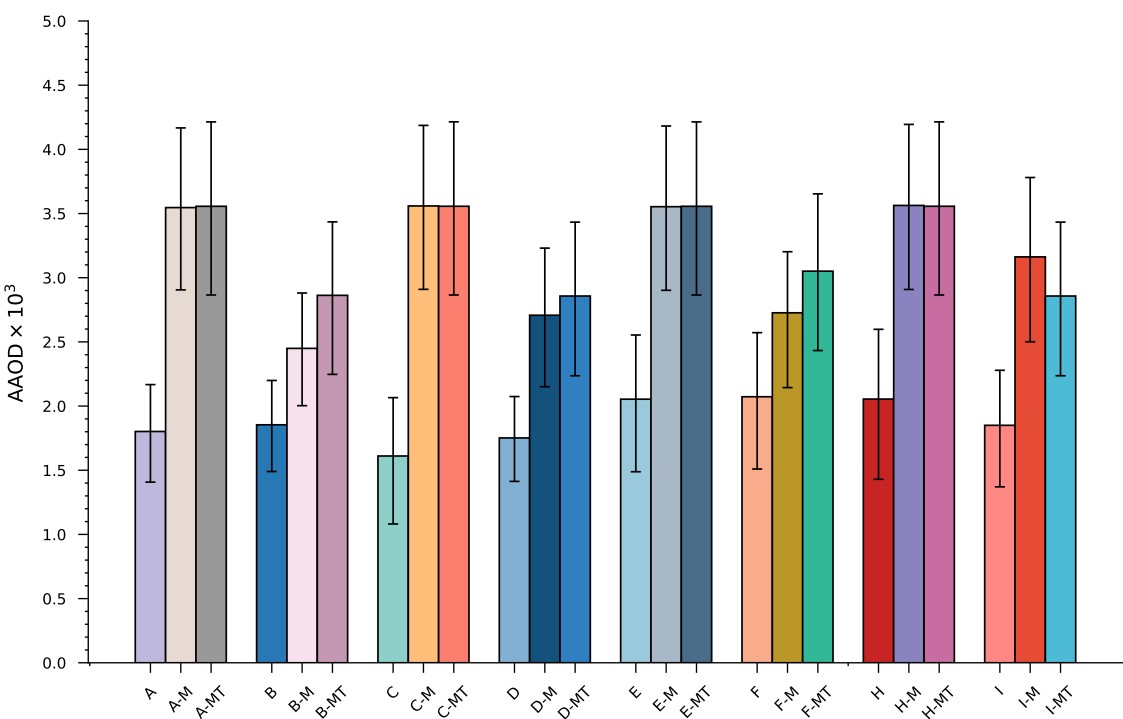

**Figure 6.** Comparing the global mean of AAOD modeled with morphologically realistic models, Mie models based on the SP2, and "true" mixing states. Mie-based models significantly overestimate the AAOD of partially-coated BC due to the assumption of fully coated BC, and the difference between Mie-based models based on SP2 retrieved and "true" $D_p/D_c$.





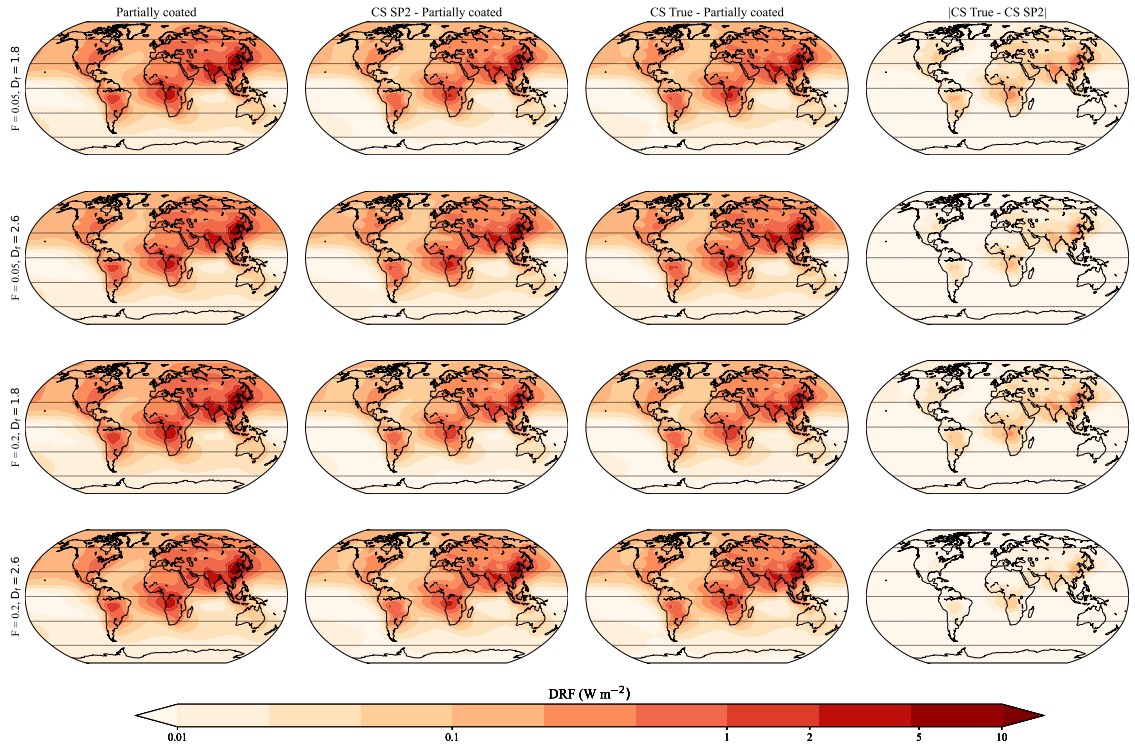

**Figure 7.** Comparisons of the global mean DRF modeled with morphologically realistic models, Mie models based on the SP2, and "true" mixing states, where $f_{BC}$ is 20%. The DRF overestimation of Mie models are even comparable to that of partially-coated BC itself. The DRF was calculated by multiplying the AAOD by $170 \pm 43$ Wm$^{-2}$ /AAOD and the contour shows the median values.





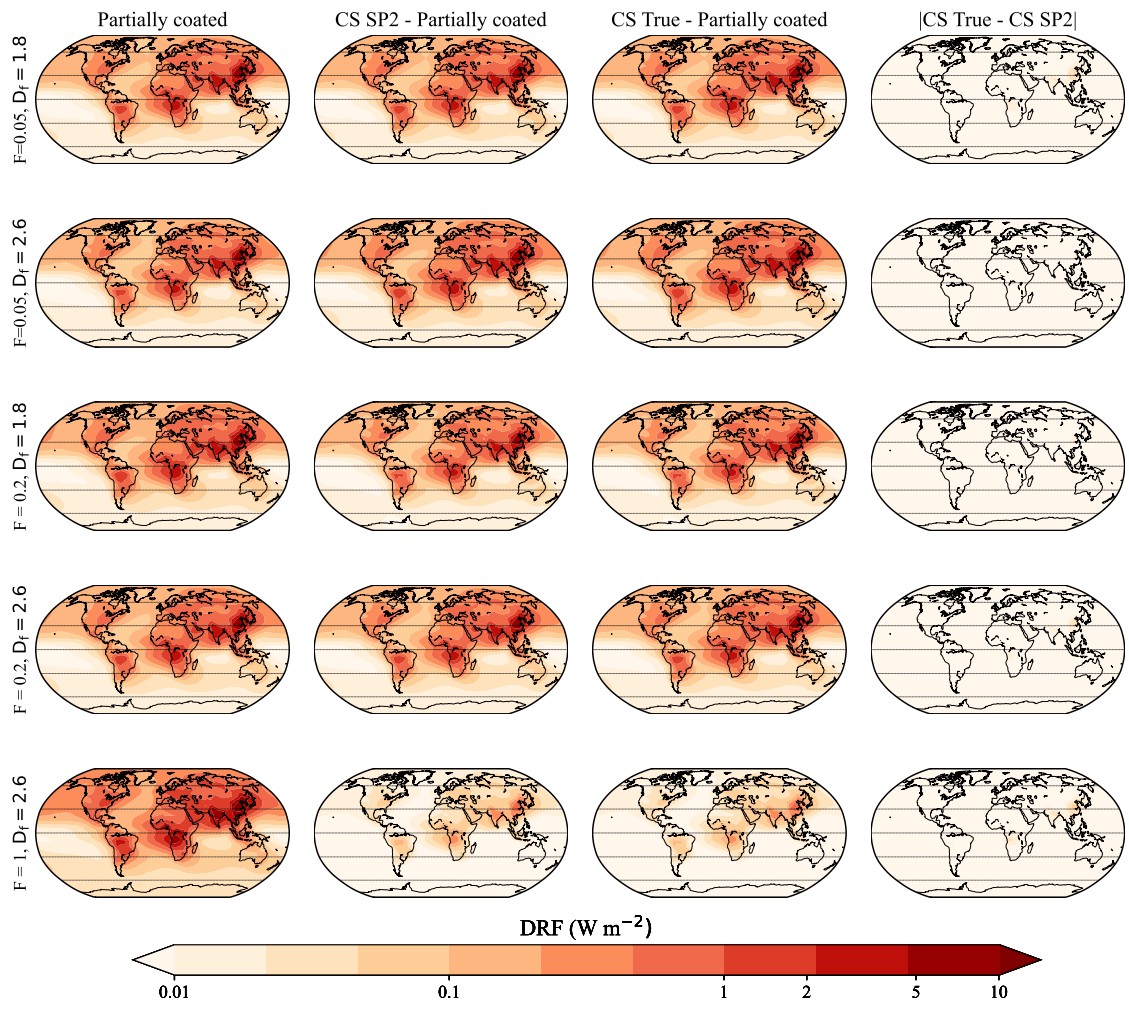

**Figure 8.** Similar to Figure 7, but for $f_{BC}$ = 10%. It can be seen that the DRF difference between Mie models based on SP2 and "True" $D_p/D_c$ is smaller than in cases where $f_{BC}$ = 20%.



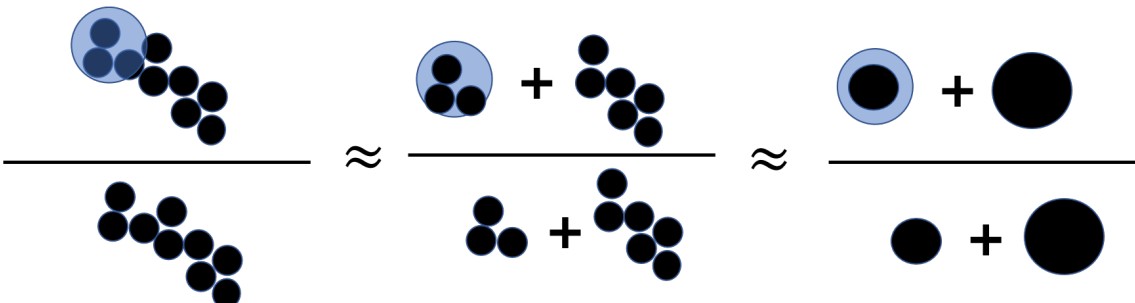

**Figure 9.** Improved Mie model by considering an additional parameter that characterizes the proportion of the coated BC core (F).

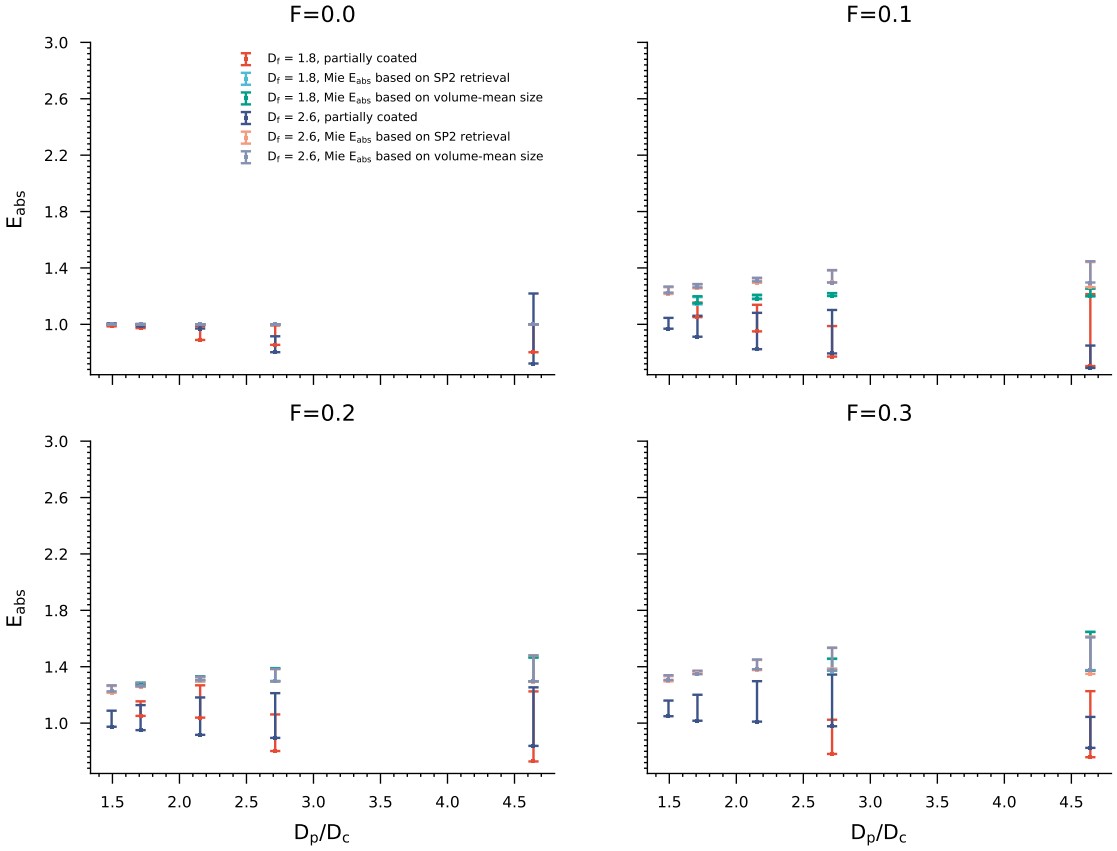

**Figure 10.** Similar to Figure 5, but taking into account the effects of F in the Mie models. The results show that the accuracy of the Mie-based $E_{abs}$ models is significantly improved when we consider external parameters to describe the effects of F.





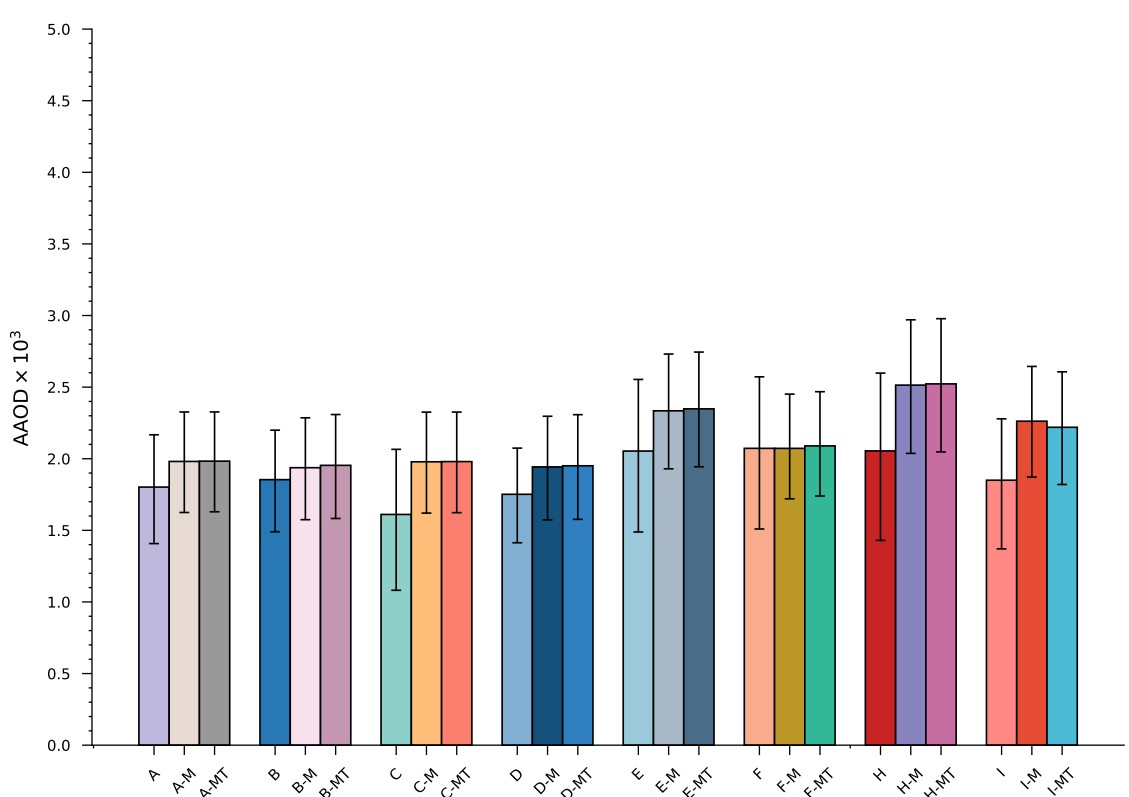

**Figure 11.** Similar to Figure 6, but taking into account the effects of F in the Mie models.



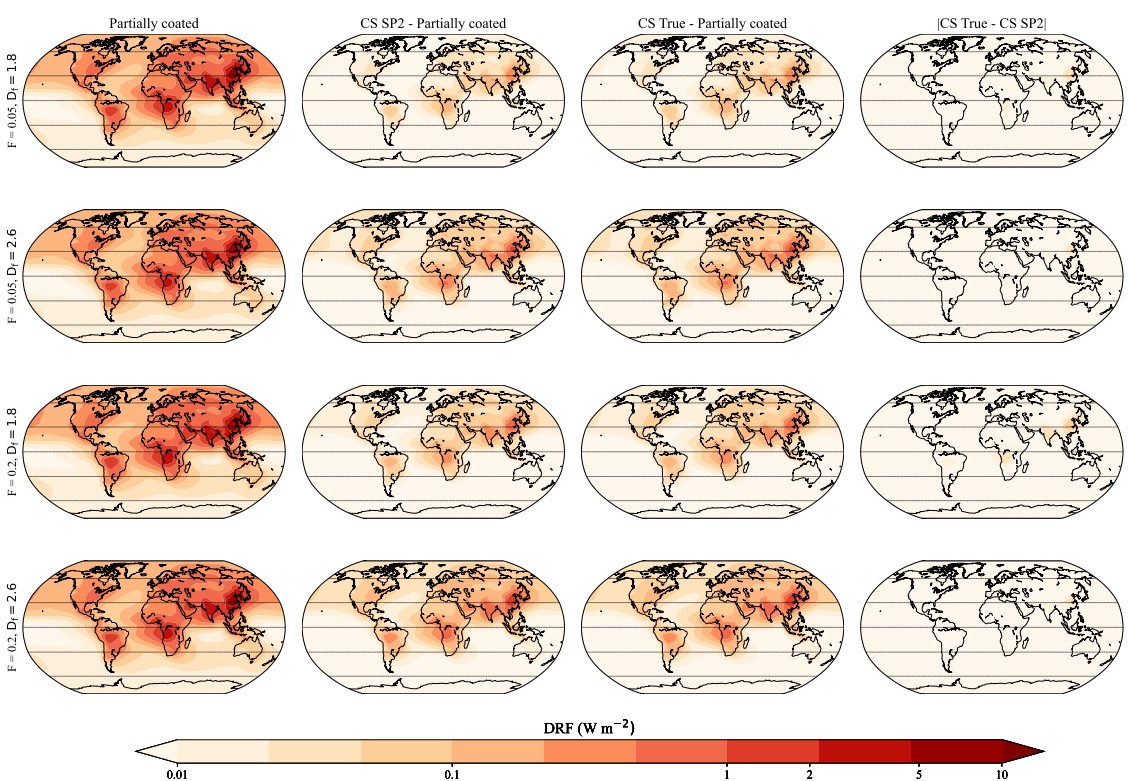

**Figure 12.** Similar to Figure 8, but taking into account the effects of F in the Mie models.



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



*Data availability.* The data can be requested from the corresponding athour.

*Author contributions.* JL and MH conceived the presented idea. JL developed the models, performed the computations, and wrote the paper. JQ, KL, HH, YS and XG verified the simulation methods and results. MH and JQ revised the paper and supervised the findings of this work. All authors discussed the results and contributed to the final paper.

*Competing interests.* The authors declare that they have no conflict of interest.

520