# Peer review of "Technical note: Numerical quantification of the mixing states of partially-coated black carbon based on the single-particle soot photometer: Implication for global radiative forcing"

_EGUsphere, 2024_

## Author Comment (AC1)

**Response to the comments of reviewers #1**

The authors are very grateful to the editors and reviewers for their valuable comments and constructive suggestions. The reviewers' questions and comments are highlighted in **BLACK font**, and the answers in **BLUE**. The changes made in the revised manuscript are highlighted in **RED**.

**Comments:** This manuscript evaluates the accuracy of core-shell Mie theory in comparison with a computationally accurate MSTM model, for fractal ($D_f$=1.8) and compact ($D_f$=2.6) aggregates with a varied coated fraction F. The authors have not directly addressed the fact that coatings will cause soot aggregates to become compact. Instead, they have cleverly included the no-restructuring and full-restructuring cases in their varying of F (the fraction of aggregate inside the coating droplet). This results in a test data set that includes the more realistic case where only the coated part of the aggregate is restructured.

Unfortunately, the manuscript has some major issues. It does not accurately acknowledge the existing literature to an extent that I have to recommend rejection for more than one reason. To fix these issues, the manuscript requires a complete rewrite, with new title, new figures, and reframing (in terms of the literature context). Therefore, it only makes sense to re-submit a new manuscript in the future.

**Response:** Thanks for your comments. We have improved the manuscript according to the constructive suggestions of you and other reviewers, and the point-to-point responses are shown in the following.

**Comments:** First, the authors have presented their work as "SP2 vesus truth" which is completely misleading. The SP2 measures scattering cross sections (line 113). The SP2 does not assume a core-shell configuration. It is SP2 users who have analyzed SP2 data using this assumption, but it is not a feature of the instrument. For example, Liu et al. (2017 http://dx.doi.org/10.1038/ngeo2901) showed that improved SP2 analysis models can be used successfully. The authors must therefore rewrite their manuscript as "core-shell vs MSTM". With this new scope, it will become obvious that the manuscript must be completely rewritten to cite and quantitatively discuss the many papers which have already discussed this topic. It needs to be clear why this manuscript should be published since many core-shell vs MSTM (or DDA) papers exist.

**Response:** Thanks very much for your comments. Indeed, SP2 itself does not assume a core-shell structure. By combining SP2 with other instruments, it is possible to obtain $D_p/D_c$ without simplifying assumptions about morphology. However, due to the high cost of these measurement instruments, many researchers do not have access to all of them simultaneously. As a result, many researchers rely on Mie scattering to invert the total particle diameter based on the scattering measured by SP2 and then calculate $D_p/D_c$. The target audience of this paper is precisely these researchers, and we acknowledge that 'SP2 versus truth' is not appropriate, so we have changed it to 'SP2-Mie versus volume-mean'. We have provided a detailed description of this in the revised manuscript.

Indeed, there exist numerous studies comparing Core-shell and MSTM models, and we have included relevant references in the revised manuscript. However, this work specifically focuses on

the inversion of $D_p/D_c$ using the Core-shell model for SP2 users, which differs from previous comparisons between Core-shell and MSTM. Here, we utilize the light scattering properties calculated by the Partially-coated model as "pseudo-measurements" and then employ Mie scattering theory to invert $D_p/D_c$, aligning with the principle of SP2 inversion for mixing states based on Mie scattering. By comparing the inverted $D_p/D_c$ with the true volumetric $D_p/D_c$, we investigate the impact of black carbon (BC) microphysical properties on the inversion of mixed states using SP2 based on Mie scattering. Wu et al. (2023) studied the influence of adopting the core-shell model on the inversion of optical particle size using SP2 based on MSTM, but their study assumed fully coated BC, neglecting partially-coated BC. However, BC is often partially coated, and the absorption of partially-coated BC is more intricate than that of fully coated BC, determined not only by the ratio of the core volume to the total particle but also by the ratio of the volume of coated BC cores to the total volume of BC cores. Furthermore, Dp/Dc has broader applications in climate research, yet Wu et al. (2023) did not explore the influence of microphysical properties on the inversion of mixed states or assess their implications for climate effects. Liu et al. (2023) attempted to use a similar model to Wu et al. (2023) to evaluate errors in mixed state due to BC morphology, but their inversion parameters differ from SP2's measurement principle, incompletely reflecting SP2's measurement process. Additionally, Liu et al. (2023) also failed to investigate the impact of partially coated BC aerosols.

Similarly, several studies have compared the differences in absorption enhancement calculated by MSTM and the core-shell model, including comparisons between the partially-coated model and the core-shell model. However, the novelty of this paper lies in evaluating the difference in absorption enhancement between the $D_p/D_c$ inverted using SP2 and that of partially-coated BC, specifically catering to SP2 users. This approach directly addresses practical SP2 measurements, differing from directly prescribed $D_p/D_c$ values, as SP2 inversions inherently contain errors. Additionally, another contribution of this study is proposing an improved Mie scattering model that considers the factor F to simulate the absorption enhancement of partially-coated BC. By incorporating both F and $D_p/D_c$ into the Mie model, we aim to enhance its calculation accuracy, significantly facilitating model applications.

Given the challenges in directly simulating F in climate models, practical applications necessitate statistical analyses of F in different regions based on observations. Nevertheless, current observation methods struggle to directly measure F, highlighting the potential of inversion methods for F estimation. In the future, SP2 can be utilized to measure $D_p/D_c$, while optical measurements can concurrently capture black carbon absorption characteristics (e.g., absorption enhancement). Leveraging the Mie scattering model presented in this paper, which accounts for F, we can retrieve F through inversion. By employing these methods to measure F under various conditions, including different regions and pollution environments, we can obtain F values tailored to those specific conditions. Ultimately, these statistically derived F values under diverse conditions can be incorporated into climate model simulations. We have clarified it in the introduction:

"The single particle soot photometer (SP2), an instrument for measuring the mass of individual BC particles, has recently been widely used to measure mixing states (Schwarz et al., 2006; R. S. Gao and Worsnop, 2007). The SP2 measures scattering of individual particles reflected from a 1064 nm laser, and the mass of the BC core is estimated from the incandescence signal (Moteki and Kondo, 2008; Wu et al., 2023). Based on an assumed BC mass density, we can calculate the mass-equivalent diameter of the BC cores. To obtain the mixing state of single-particle black carbon (BC), many

researchers have attempted to develop methods for simultaneously measuring the total particle size online. A significant advancement in this field has been achieved by combining SP2 with the Centrifugal Particle Mass Analyzer (CPMA) (Olfert and Collings, 2005; Liu et al., 2017b; Yu et al., 2020; Naseri et al., 2024). In this process, CPMA measures the mass of individual particles to infer particle size, without simplifying morphological features. Some techniques based on the differential mobility analyzer (DMA)-SP2 system was also developed (Andrew R. Metcalf and Seinfeld, 2013; Zhao et al., 2022; Huang et al., 2024). However, due to the high cost of these instruments, many researchers find it challenging to own them simultaneously. In addition, in some cases (such as unmanned aerial vehicle detection), it is very inconvenient to measure by combining so many instruments. Consequently, alternative methods for measuring mixing states have been adopted (R. S. Gao and Worsnop, 2007; Naseri et al., 2024; Lee et al., 2022). The SP2 can simultaneously provide information on the scattering properties of particles, and the leading-edge-only (LEO) technique enables the extraction of particle scattering signals. Many researchers thus invert the total particle size based on Mie theory, utilizing the scattering signals measured by the SP2 (R. S. Gao and Worsnop, 2007). Nevertheless, this approach has limitations: coated BC often does not exhibit a perfect core-shell structure. Despite the limitations of instruments and costs, this method, subsequently referred to as SP2-Mie, is still widely used by researchers. A primary target audience of this paper is researchers who employ the SP2 for measuring BC mixing states based on Mie scattering. Nevertheless, current research on explaining the uncertainties associated with the SP2-Mie method remains limited.

SP2 users often explain the measured mixed state with Mie scattering (Moteki and Kondo, 2008; Schwarz et al., 2008; Naseri et al., 2024). However, Moteki et al. (2014) found that the discrepancy between the calculated results of Mie scattering and the scattering cross-section measured by SP2 can reach up to 40% in some cases. One of the important reasons is that the morphology of BC is often complex and frequently partially-coated (Adachi et al., 2007; China et al., 2013; Wang et al., 2017). Although previous studies have recognized that simplifying the microphysical properties of BC aerosols can lead to inaccurate determination of mixing states (Schwarz et al., 2015), there is still a lack of quantification of the effects of microphysical properties.

Previous studies have compared the scattering cross-section of core-shell BC and more morphologically realistic BC and found that the scattering properties of BC are significantly influenced by morphologies (Kahnert et al., 2013; Scarnato et al., 2013; Luo et al., 2019; Kahnert and Kanngießer, 2021). However, direct comparison of mixing states determined based on SP2-Mie and volume mean is very limited. In a recent study by Wu et al. (2023), the effects of adopting the core-shell model on the inversion of optical particle size using the SP2 based on the multiple sphere T-matrix (MSTM) were investigated, but they assumed fully coated BC and neglected partially-coated BC. However, partially-coated BC is often partially-coated, and the absorption of partially-coated BC is more complex than that of fully coated BC. It is determined not only by the ratio of the core size to the total particle ($D_p/D_c$), but also by the ratio of the volume of the coated BC cores to the total volume of the BC cores (F). In addition, $D_p/D_c$ has a broader application in climate research, but Wu et al. (2023) did not investigate the influence of microphysical properties on the inversion of mixed states or evaluate their impact on climate effects. Liu et al. (2023) attempted to use a similar model to Wu et al. (2023) to evaluate mixed-state errors due to BC morphology, but their inversion parameters are based on the differential scattering cross section, which is different

from the measurement principle of the SP2 and only incompletely reflects the measurement process of the SP2. The scattering signal measured by the SP2 should be proportional to the scattering cross section within the measurement angle range, rather than the differential scattering cross section (Moteki and Kondo, 2008; Wu et al., 2023; Naseri et al., 2024). In addition, the effect of partially-coated BC aerosols was not investigated in Liu et al. (2023). The aim of this work is not to discredit the use of the SP2 to measure mixture states, but rather to theoretically investigate the inffuence of the microphysical properties of BC on the accuracy of the SP2-Mie method and to assist researchers in analyzing the sources of measurement uncertainty of the SP2-Mie method during actual measurements.

Another important concern for SP2 users is the absorption enhancement of coated BC. When BC is mixed with other components, its total absorption can be enhanced due to the "lensing effect." In reality, the mixing state of BC significantly influences absorption enhancement, making the mixing state measured by SP2 crucial for absorption enhancement calculations and climate predictions. However, as mentioned above, when using the mixing state measured by SP2 to calculate absorption enhancement, a core-shell structure is commonly assumed and Mie scattering calculations are used. Another objective of this work is to evaluate the uncertainties in calculating absorption enhancement using the SP2-Mie determined mixing state and to investigate methods for improvement. Previous researchers have conducted a number of studies comparing the absorption enhancement of BC with complex morphology and their Mie scattering results, including studies on partially-coated BC (Zhang et al., 2017, 2018; Luo et al., 2018; Wang et al., 2021b). However, an evaluation specifically for SP2-Mie users is lacking. In practical measurements, the mixing state measured by SP2-Mie is often used to calculate the absorption enhancement, which may differ from the volume-mean mixing state. Therefore, it is necessary to evaluate the mixing condition determined by SP2-Mie. In addition, previous studies have shown that the absorption gain of partially-coated BC is simultaneously affected by F and $D_p/D_c$. However, in real situations, it is difficult to obtain F with realistic morphology models. Developing a simplified model that accounts for both F and $D_p/D_c$ is important for determining F during measurements and for statistically analyzing F in different regions, thereby improving the accuracy of climate simulations."

**Comments:** Second, the fact that the relative position of the coating and core has a huge influence on absorption properties was also shown in many other studies. For example, by Fuller et al. (1999) and multiple others. This manuscript's approach to studying this topic might be a worthwhile contribution to the existing literature. But the authors have to show this. They have to cite and discuss those previous manuscripts if they believe their contribution adds to them. The authors' observation that partially coated aggregates can be treated as a sum of coated and uncoated particles was interesting and should be explored. Again, this would require completely rewriting the manuscript. Also, this would require the authors to study values of F from 0.0 to 1.0, and not only to 0.3. There is no reason to stop at 0.3.

**Response:** Thanks for your comments.  As shown in above, the novelty of this paper lies in evaluating the difference in absorption enhancement between the $D_p/Dc$ inverted using SP2 and that of partially-coated BC, specifically catering to SP2 users. This approach directly addresses practical SP2 measurements, differing from directly prescribed $D_p/D_c$ values, as SP2 inversions inherently contain errors. Additionally, another contribution of this study is proposing an improved Mie scattering model that considers the factor F to simulate the absorption enhancement of partiallycoated BC. By incorporating both F and Dp/Dc into the Mie model, we aim to enhance its calculation accuracy, significantly facilitating model applications.

Due to the assumption of spherical coating in this paper, for fluffy black carbon cores, when the coating thickness is small and F is large, it becomes difficult to find a corresponding coating position. Therefore, for fluffy black carbon, only variations of F from 0 to 0.3 were considered, which is also a limitation acknowledged in the paper. Taking into account your insightful comments, we have extended the range of F to vary from 0 to 1 in the case of compact black carbon cores, and some results are placed in the support information. We have clarified this aspect and added atmospheric implications in the revised manuscript. Please refer to the revised document for detailed information.

To illustrate the importance of the simplified Mie model developed in this paper for future climate simulations, we have added the following description in the "Atmospheric Implication" section:

"Since it is difficult to simulate F directly in climate models, a statistical analysis of F in different regions by observations is required for practical applications. However, current observational methods also have difficulty in observing F directly, so inversion methods can be used for F measurement. In the future, SP2 can be used to measure Dp/Dc, and optical measurements can also be used to obtain BC absorption properties (such as absorption enhancement). With the Mie scattering model proposed in this paper, which takes F into account, F can be determined by inversion. By applying such methods to measure F under different conditions, including different regions and pollution environments, we can obtain F values under different conditions. Finally, these statistical F values under different conditions can be used in climate model simulations. Recent studies have shown that Mie-based estimates of absorption enhancement can still overestimate measured values even when accounting for non-uniform mixing states, underlining the importance of our proposed refinement (Huang et al., 2024; Fierce et al., 2020)."

**Comments:** Finally, the use of a global chemical transport model in the present study to illustrate the effects of the soot MAC introduced an unnecessary and distracting complexity. No parameters except MAC were varied in the model. Therefore, simply plot MAC. Using an extremely complex model to demonstrate a minor point (influence of F) does not provide scientific insight.

**Response:** Thank you for your comment. In this paper, indeed, there is no other influence besides MAC. In fact, apart from absorption enhancement, no other parameters have been altered, and the influence of the absorption by the black carbon core itself has been neglected. This was set based on the assumption that users of SP2 are primarily concerned with the impact of absorption enhancement. The mass absorption cross-section (MAC) of bare black carbon is indeed affected by its morphology, but numerous studies have conducted extensive measurements and simulation research on this topic, leading to a clearer understanding of its uncertainty. This current research primarily focuses on the understanding of the impact of mixing characteristics on SP2 users' measurements. Since mixing characteristics mainly affect absorption enhancement, this study primarily concentrates on the influence of absorption enhancement. For the effects of morphology on the black carbon core, please refer to other literature. Furthermore, in the process of model application, due to the current underestimation of model predictions for the MAC of bare black carbon, many researchers in climate modeling studies have adopted a MAC value of 7.5 m²/g, as it is derived from measurements. To focus on SP2 users and the impact of black carbon absorption enhancement, this paper adopts a black carbon core MAC of 7.5 m²/g. However, to illustrate the influence of black carbon core morphology on MAC, we have included in the appendix the results

of the impact of different black carbon core morphologies on black carbon MAC.

However, considering that some researchers in climate studies who conduct uncertainty analyses may be interested in the large-scale absorption and climate change impacts caused by these micro-scale properties, we have retained the calculations from our atmospheric chemical transport model. However, considering that your opinion is very reasonable, we only keep the bar chart of the global average AAOD, and the spatial distribution map of DRF is moved to the support information for reference only. Indeed, our calculations involve many simplifications, such as our assumption that all BC is partially-coated, whereas in reality, there exists a variety of BC particles, and each BC particle has a unique Dp/Dc ratio. Nevertheless, the sensitivity calculations in this paper aim to emphasize the impact of partially-coated BC and estimate the upper limit of potential errors. This may provide some assistance to researchers investigating uncertainties in climate simulations, and therefore, we have retained this section of the content.

---

## Author Comment (AC2)

**Response to the comments of reviewers #2**

The authors are very grateful to the editors and reviewers for their valuable comments and constructive suggestions. The reviewers' questions and comments are highlighted in **BLACK font**, and the answers in **BLUE**. The changes made in the revised manuscript are highlighted in **RED**.

**Comments:** This manuscript quantifies the uncertainty of the mixing state and absorption of partially-coated BC based on SP2 with numerical simulation, and points out that quantification of the mixing states of partially-coated black carbon based on the single-particle soot, as well as its impacts on BC absorption and radiative forcing. Based on the simulation, the authors suggest adding a parameter F to model the radiative effect of BC in climate modeling. In general, the manuscript is not rigorously organized and the figures are not clear. The following issues should be taken into consideration for improvement.

The manuscript highlights the importance of partially-coated BC, and there is large uncertainty when assuming BC is fully coated. The title is "Numerical quantification of the mixing states of partially-coated black carbon based on the single-particle soot photometer", but there is no measurement data from SP2. In fact, it is just the difference between Mie theory and MSTM. This problem is common in current regional or climate model, not just for SP2. Zhang et al (2018) did similar work on numerical simulation of partially-coated BC absorption. In the introduction part, the progress of studying on partially-coated BC should be summarized and the novelty of this study should be pointed out.

**Response:** Thanks very much for your comments. There is indeed no SP2 measurements. We combined the calculations from the MSTM and the SP2 measurement principle to represents the "pseudo measurement", and then retrieved the mixing states based on the Mie theory like many SP2 user done. By exploring the effects of the microphysical properties on the calculations, we can provide some insights on the uncertainties of the mixing states based on the SP2 measurement and Mie retrievals, and this is one of the main difference from Zhang et al. (2018).

Indeed, like Reviewer #1 says, SP2 just detects scattering, does not assume core-shell structure, and some techniques that don't use the core-shell Mie theory was also proposed. However, these techniques commonly needs the combinations of different instruments, and it's difficult for many people to own so many expensive instruments at the same time. Therefore, The measurement based on SP2 using Mie scattering to retrieve the mixing state of black carbon remains a preferred choice for many researchers. This study primarily focuses on SP2 users who employ Mie scattering for measuring the mixing state, and we have clarified this in the revised manuscript.

In addition, another distinction from Zhang et al. (2018) is the exploration of using multiple spheres to improve the calculation of absorption enhancement via Mie scattering. This is also a significant demand for many SP2 users when measuring the mixing state. While the uncertainty of bare black carbon absorption has been relatively well studied through the efforts of many researchers, the mixing state can significantly affect the absorption enhancement of black carbon. One of the main purposes of SP2 mixing state measurement is to reduce the uncertainty of absorption enhancement through the understanding of particle mixing states. For most SP2 users, Mie scattering is often used to calculate the absorption enhancement of particle ensembles after measuring Dp/Dc, while

ignoring the influence of F. Zhang et al. (2018) demonstrated through the Multi-Sphere T-Matrix (MSTM) method that F can affect absorption enhancement. However, it is difficult to apply an MSTM model to SP2 measurements and climate models due to the variations in F among individual particles.

Another contribution of this paper is to explore the improvement in absorption enhancement calculations by considering the influence of F based on Mie scattering. This is a highly application-oriented approach. If Mie scattering is feasible, we can simultaneously retrieve F and Dp/Dc based on SP2 measurements and absorption measurements, obtaining F and Dp/Dc values in different regions. This will further constrain the radiative effects of black carbon at the single-particle level, which has significant practical value for the future. We will also conduct further research in the near future. We have re-written the introduction in the revised manuscript:

"The single particle soot photometer (SP2), an instrument for measuring the mass of individual BC particles, has recently been widely used to measure mixing states (Schwarz et al., 2006; R. S. Gao and Worsnop, 2007). The SP2 measures scattering of individual particles reflected from a 1064 nm laser, and the mass of the BC core is estimated from the incandescence signal (Moteki and Kondo, 2008; Wu et al., 2023). Based on an assumed BC mass density, we can calculate the mass-equivalent diameter of the BC cores. To obtain the mixing state of single-particle black carbon (BC), many researchers have attempted to develop methods for simultaneously measuring the total particle size online. A significant advancement in this field has been achieved by combining SP2 with the Centrifugal Particle Mass Analyzer (CPMA) (Olfert and Collings, 2005; Liu et al., 2017b; Yu et al., 2020; Naseri et al., 2024). In this process, CPMA measures the mass of individual particles to infer particle size, without simplifying morphological features. Some techniques based on the differential mobility analyzer (DMA)-SP2 system was also developed (Andrew R. Metcalf and Seinfeld, 2013; Zhao et al., 2022; Huang et al., 2024). However, due to the high cost of these instruments, many researchers find it challenging to own them simultaneously. In addition, in some cases (such as unmanned aerial vehicle detection), it is very inconvenient to measure by combining so many instruments. Consequently, alternative methods for measuring mixing states have been adopted (R. S. Gao and Worsnop, 2007; Naseri et al., 2024; Lee et al., 2022). The SP2 can simultaneously provide information on the scattering properties of particles, and the leading-edge-only (LEO) technique enables the extraction of particle scattering signals. Many researchers thus invert the total particle size based on Mie theory, utilizing the scattering signals measured by the SP2 (R. S. Gao and Worsnop, 2007). Nevertheless, this approach has limitations: coated BC often does not exhibit a perfect core-shell structure. Despite the limitations of instruments and costs, this method, subsequently referred to as SP2-Mie, is still widely used by researchers. A primary target audience of this paper is researchers who employ the SP2 for measuring BC mixing states based on Mie scattering. Nevertheless, current research on explaining the uncertainties associated with the SP2-Mie method remains limited.

SP2 users often explain the measured mixed state with Mie scattering (Moteki and Kondo, 2008; Schwarz et al., 2008; Naseri et al., 2024). However, Moteki et al. (2014) found that the discrepancy between the calculated results of Mie scattering and the scattering cross-section measured by SP2 can reach up to 40% in some cases. One of the important reasons is that the morphology of BC is often complex and frequently partially-coated (Adachi et al., 2007; China et al., 2013; Wang et al., 2017). Although previous studies have recognized that simplifying the microphysical properties

of BC aerosols can lead to inaccurate determination of mixing states (Schwarz et al., 2015), there is still a lack of quantification of the effects of microphysical properties.

Previous studies have compared the scattering cross-section of core-shell BC and more morphologically realistic BC, and they found that the scattering properties of BC is significantly affected by morphologies (Schwarz et al., 2008). However, the direct comparison of the mixing states retrieved based on the SP2-Mie and the volume-mean are very limited. A recent study by Wu et al. (2023) has studied the impact of adopting the core-shell model on the inversion of optical particle size using the SP2 based on the the multiple-sphere T-matrix (MSTM), but their study assumed fully coated BC, neglecting partially coated BC. However, BC is often partially coated, and the absorption of partially coated BC is more complex than that of fully coated BC, determined not only by the ratio of the core size to the total particle ($D_p/D_c$) but also by the ratio of the volume of coated BC cores to the total volume of BC cores (F). Furthermore, $D_p/D_c$ has broader applications in climate research, yet Wu et al. (2023) did not explore the influence of microphysical properties on the inversion of mixed states or assess their implications for climate effects. Liu et al. (2023) attempted to use a similar model to Wu et al. (2023) to evaluate errors in the mixed state due to BC morphology, but their inversion parameters are based on the differential scattering cross section which differ from the measurement principle of the SP2, incompletely reflecting the SP2's measurement process. The scattered signal measured by SP2 should be proportional to the scattering cross-section within the measurement angle range, rather than the differential scattering cross section (Moteki and Kondo, 2008; Wu et al., 2023; Naseri et al., 2024) Additionally, Liu et al. (2023) also failed to investigate the impact of partially coated BC aerosols. The aim of this paper is not to discredit the use of the SP2 for measuring mixing states but rather to theoretically investigate the influence of BC microphysical properties on the accuracy of the SP2-Mie method, assisting researchers in analyzing the sources of measurement uncertainty in the SP2-Mie method during actual measurements.

Another important concern for SP2 users is the absorption enhancement of coated BC. When BC is mixed with other components, its total absorption can be enhanced due to the "lensing effect." In reality, the mixing state of BC significantly influences absorption enhancement, making the mixing state measured by SP2 crucial for absorption enhancement calculations and climate predictions. However, as mentioned above, when using the mixing state measured by SP2 to calculate absorption enhancement, it is still common to assume a core-shell structure and use Mie scattering calculations. Another objective of this paper is to assess the uncertainties in calculating absorption enhancement using the mixing state retrieved by SP2-Mie and to explore methods for improvement. Previous researchers have conducted a series of studies comparing the absorption enhancement of BC with complex morphologies and their Mie scattering results, including studies on partially-coated BC (Wang et al., 2021b). However, there is a lack of evaluation specifically for SP2-Mie users. In practical measurements, the mixing state measured by SP2-Mie is often used to calculate absorption enhancement, which may differ from the "true" mixing state. Therefore, it is necessary to evaluate the mixing state retrieved by SP2-Mie. Furthermore, previous studies have found that the absorption enhancement of partially-coated BC is simultaneously influenced by both F and Dp/Dc. However, in real-world situations, it is difficult to obtain F using realistic morphology models. Developing a simplified model that considers both F and Dp/Dc is significant for retrieving F during

measurements and for statistical analysis of F in different regions, thereby improving the accuracy of climate simulations.”

Zhang, X., Mao, M., Yin, Y., and Wang, B.: Numerical investigation on absorption enhancement of black carbon aerosols partially coated with nonabsorbing organics, Journal of Geophysical Research: Atmospheres, 123, 1297–1308, https://doi.org/https://doi.org/10.1002/2017JD027833, 2018.

**Comments:** As stated in the manuscript, the model calculated MACBC is inconsistent with the measured MACBC, and most models underestimate $MAC_{BC}$ based on the measured mass density and refractive index, why did the authors choose a MACBC of 7.5 m$^2$g−1 in this study? Both fluffy and compact BC aggregates were considered in this study, and MACBC also varies for both BC shapes. How does MACBC affect the calculated absorption and radiative forcing?

**Response:** Thanks for your comments. The mass absorption cross-section (MAC) of bare black carbon is indeed affected by its morphology, but numerous studies have conducted extensive measurements and simulation research on this topic, leading to a clearer understanding of its uncertainty. This current research primarily focuses on the understanding of the impact of mixing characteristics on SP2 users' measurements. Since mixing characteristics mainly affect absorption enhancement, this study primarily concentrates on the influence of absorption enhancement. For the effects of morphology on the black carbon core, please refer to other literature. Furthermore, in the process of model application, due to the current underestimation of model predictions for the MAC of bare black carbon, many researchers in climate modeling studies have adopted a MAC value of 7.5 m²/g, as it is derived from measurements. To focus on SP2 users and the impact of black carbon absorption enhancement, this paper adopts a black carbon core MAC of 7.5 m²/g. We have made clarifications in the revised manuscript:

“It should be noted that in this work we assume that the mass absorption cross-section (MAC) of the BC core is fixed. In reality, however, the MAC of the BC core is also influenced by its morphology. Nevertheless, this study primarily aims to understand the influence of mixture properties on the measurements performed by SP2 users. Since mixing states primarily affect absorption enhancement, we focus primarily on the effects of absorption enhancement. Besides, many climate modeling studies have also used a fixed MAC from measurements because model predictions for the MAC of BC core are currently underestimated, and the total absorption is estimated by multiplying the absorption enhancement (Bey et al., 2001; Eastham et al., 2018; Wang et al., 2014; Zhang et al., 2021). In this process, the absorption enhancement of BC is the main influencing parameter (Wang et al., 2014; Zhang et al., 2021). To focus on SP2 users and the effects of BC absorption enhancement, a BC core MAC of 7.5 ± 1.2 m$^2$g$^{-1}$ is assumed in this work. Regarding the influence of morphology on the MAC of the BC core, previous studies have performed extensive measurements and simulation studies that provide a clearer picture of the uncertainties (Liu and Mishchenko, 2005; Kahnert, 2010; Luo et al., 2018; Fengshan Liu and Corbin, 2020). Our study is therefore primarily concerned with the effects on absorption enhancement.”

**Comments:** In the fourth paragraph, it is mentioned twice that “BC is often partially-coated”. If it was written carefully, I think the authors want to emphasize the importance of partially-coated BC. However, the exact fraction of partially-coated BC in the real atmosphere is more convincing than current expression. In addition, the fraction of partially-coated BC in the real atmosphere also impacts on the uncertainties of climate model, so the uncertainty of climate model is not reliable when consider partially-coated BC alone.

**Response:** Thanks very much for your comments. Thank you very much for your comment. Indeed, in order to conduct a more accurate assessment, we should consider the proportion of partially covered black carbon. However, the calculations in this paper are merely for sensitivity analysis, aiming to demonstrate the significance of the microphysical properties of partially covered black carbon in global climate simulations. More precise simulations in the future will need to take into account the proportions of BC with different morphologies. We have described this in the results of the revised manuscript:

"It is worth noting that all the BC considered in this study is assumed to be partially-coated, while in reality, there exist various types of BC. The sensitivity analysis conducted in this study is solely to illustrate the importance of the microphysical properties of partially covered BC in SP2-Mie inversion and global climate assessment. For more accurate climate simulations in the future, it will be necessary to consider the proportions of BC with different morphologies."

However, considering that your opinion is very reasonable, we only keep the bar chart of the global average AAOD, and the spatial distribution map of DRF is moved to the appendix for reference only.

**Comments:** The authors propose to consider not only the effects of mixing states (Dp/Dc) but also the effects of the proportion of the coated BC core (F) in climate model. So how to determine F value in climate model? Any suggestions?

**Response:** Thanks for your comments. This paper discovers that in addition to $D_p/D_c$, F also significantly impacts climate predictions. Since it is challenging to directly simulate F in climate models, statistical analysis of F in different regions through observations is required for practical applications. However, current observation methods also face difficulties in directly observing F, so inversion methods can be leveraged for F measurement. In the future, SP2 can be used to measure $D_p/D_c$, and optical measurements can also be employed to obtain black carbon absorption characteristics (such as absorption enhancement). With the Mie scattering model proposed in this paper that considers F, F can be retrieved through inversion. By adopting such methods to measure F under various conditions, including different regions and pollution environments, we can obtain F values under different conditions. Finally, these F statistical values under different conditions can be utilized in climate model simulations. We have clarified this in the revised manuscript:

"Since it is difficult to simulate F directly in climate models, a statistical analysis of F in different regions by observations is required for practical applications. However, current observational methods also have difficulty in observing F directly, so inversion methods can be used for F measurement. In the future, SP2 can be used to measure $D_p/D_c$, and optical measurements can also be used to obtain BC absorption properties (such as absorption enhancement). With the Mie scattering model proposed in this paper, which takes F into account, F can be determined by inversion. By applying such methods to measure F under different conditions, including different regions and pollution environments, we can obtain F values under different conditions. Finally, these statistical F values under different conditions can be used in climate model simulations. Recent studies have shown that Mie-based estimates of absorption enhancement can still overestimate measured values even when accounting for non-uniform mixing states, underlining the importance of our proposed refinement (Huang et al., 2024; Fierce et al., 2020)."

**Comments:** Lines 39: please add the corresponding references that point out that that simplifying the microphysical properties of BC aerosols can lead to inaccurate determination of mixing states.

**Response:** Thanks for your comments. We have re-written this paragraph and added some references in the revised manuscript.

**Comments:** Line 75: the second "of" should be changed as "and".

**Response:** Thanks for your comments. We have corrected it in the revised manuscript.

**Comments:** The font size varies a lot in different figures. For figure 2, 5, 6, 7,10, 11 and 12, the font size should be enlarged.

**Response:** Thanks for your comments and suggestions. We have enlarged the font size in the revised manuscript.

**Comments:** In figure 2, 5 and 10, it is hard to figure out the results, because there are too many legends and the colors are hard to distinguish. Please replot and improve the quality.

**Response:** Thanks for your comments and suggestions. To make the figures clearer, we have divided one figure to different figures in the revised manuscript.

**Comments:** What is the difference between "BC" and "BCs"? Why do the authors use these two expressions?

**Response:** Thanks for your comments. All "BCs" are uniformly expressed as "BC" in the revised manuscript.